# Adaptive Capacities for Diversified Flood Risk Management Strategies: Learning from Pilot Projects

**Flavia Simona Cosoveanu \*, Jean-Marie Buijs \*, Marloes Bakker and Teun Terpstra**

Department of Technology, Water & Environment; HZ University of Applied Sciences, Het Groene Woud 1, 4331 NB Middelburg, The Netherlands; m.h.n.bakker@hz.nl (M.B.); t.terpstra@hz.nl (T.T.)

**\*** Correspondence: coso0002@hz.nl (F.S.C.); jm.buijs@hz.nl (J.-M.B.)

**Abstract:** Diversification of flood risk management strategies (FRMS) in response to climate change relies on the adaptive capacities of institutions. Although adaptive capacities enable flexibility and adjustment, more empirical research is needed to better grasp the role of adaptive capacities to accommodate expected climate change effects. This paper presents an analytical framework based on the Adaptive Capacity Wheel (ACW) and Triple-loop Learning. The framework is applied to evaluate the adaptive capacities that were missing, employed, and developed throughout the 'Alblasserwaard-Vijfheerenlanden' (The Netherlands) and the 'Wesermarsch' (Germany) pilot projects. Evaluations were performed using questionnaires, interviews, and focus groups. From the 22 capacities of ACW, three capacities were identified important for diversifying the current FRMS; the capacity to develop a greater variety of solutions, continuous access to information about diversified FRMS, and collaborative leadership. Hardly any capacities related to 'learning' and 'governance' were mentioned by the stakeholders. From a further reflection on the data, we inferred that the pilot projects performed single-loop learning (incremental learning: 'are we doing what we do right?'), rather than double-loop learning (reframing: 'are we doing the right things?'). As the development of the framework is part of ongoing research, some directions for improvement are highlighted.

**Keywords:** adaptive capacities; diversified flood risk management strategies; pilot project; governance networks; learning

## 1. Introduction

Climate change will result in increased exposure of low-lying coastal areas to risks associated with sea level rise. Human and ecological systems will be faced with increased saltwater intrusion, flooding, and damage to infrastructure [1]. In response to the increasing risks, EU countries are making efforts to diversify their flood risk management strategies (FRMS) by combining flood risk measures spanning the whole disaster risk management cycle (pro-action, protection, mitigation, preparation, and recovery). This includes dyke reinforcements, compartments, flood proof houses, retention areas, and crisis management [2–5]. The diversification of FRMS has enabled more options for flexibility and adaptability of flood risk management [3]. The potential for diversification of strategies (e.g., emergency plans, zoning of flood prone areas, etc.) in response to climate change depends on the adaptation space and capacity of institutions [6]. Adaptive capacities enable a flexible response, learning, and adjustment by governance networks [7–10]. However, there is a gap of theoretical formulations that connects adaptive capacity and adaptation outcomes [11], such as the role of adaptive capacities to accommodate expected climate change effects [12].

In addition, more empirical research is needed to learn from ongoing attempts to diversify FRMS and investigate how governance challenges are addressed [4]. In particular, a better understanding of the required governance arrangements and how these are formed is lacking [13]. Transforming

existing or forming new arrangements depends upon a larger variety of skills and capabilities of governance networks [14]. Herein, governance networks are defined as 'the set of conscious steering attempts or strategies of actors within governance networks aimed at influencing interaction processes and/or the characteristics of these networks'. When applying these strategies, actors produce outcomes such as changes or new solutions, policies, or services [15] (p. 11). Working towards these changes/transformations in FRM is typically done in pilot projects which provide an opportunity for experimenting and learning [16,17]. Knowledge and experiences acquired from pilot projects are often valuable lessons for upscaling of pilot project results [16,18]. Moreover, by learning from new insights and experiences, actors foster their capacities in governance networks to cope with uncertainty and change [14].

This research work attempts to fill these knowledge gaps by analysing the role of adaptive capacities for the development and implementation of diversified FRMS. The study was conducted by analysing the adaptive capacities of governance networks in two pilot projects in The Netherlands and Germany to better understand the required adaptive capacities of governance networks for implementing more diversified FRMS. The two pilot projects in this study are the 'Alblasserwaard-Vijfheerenlanden' located in the downstream area of the river Rhine in The Netherlands, and the 'Wesermarsch' located at the German coast. Both pilot projects are part of the EU Interreg project Flood Resilient Areas by Multi-layered Safety (FRAMES, [19]). Traditionally, both Dutch and German FRM is primarily based on flood protection through dykes and barriers. In addition, both countries are working on further diversification of FRMS by investing in preparedness. This includes the development of evacuation strategies, raising risk awareness, and stimulating preparedness among citizens.

The following sections present the theoretical framework (Section 2), research questions (Section 3), and the methodology for analyzing adaptive capacities (Section 4). Sections 5–7 present the results, the discussion of the outcomes, and the conclusion of this research, respectively.

## 2. Theoretical Framework

The theoretical framework aims to understand and learn from pilot projects regarding capacity development in the transition towards more diversified FRMS. Pilot projects are spaces where learning processes occur [20,21], and new insights arise related to transition dynamics [20]. The theoretical framework is grounded in the Adaptive Capacity Wheel (ACW) [22] and the Triple-loop Learning approach [23].

### 2.1. The Adaptive Capacity Wheel

The ACW is an analytical framework that enables the evaluation of institutions' ability to foster the adaptive capacity of governance networks over time and across scales. It defines six main dimensions divided into 22 adaptive capacities (Table 1). The ACW has been applied to a myriad of water related topics, including climate change [24], droughts and floods [25], sustainability of water governance systems [26,27], and policy content analysis [28] but has not yet been applied to the diversification of FRM.

Previous studies, dealing with adaptive capacities, encountered various challenges rooted in understanding the capacities' definitions [25,28], and weighting can lead to misinterpretations [27]. Because of the downsides encountered by scholars, for this study, the ACW is applied only as a qualitative method, not as a quantitative or measurable approach.

**Table 1.** Dimensions and adaptive capacities definitions [22].

| Adaptive Capacities | Definitions |
| --- | --- |
| **1. Variety** | |
| 1.1 Variety of problem frames | Room for multiple frames of references, opinions, and problem definitions |
| 1.2 Multi-actor, multi-level, multi-sector (stakeholders) | Involvement of different actors, levels and sectors in the governance process |
| 1.3 Diversity of solutions | Availability of a wide range of different solutions/pathways/actions to tackle a problem |
| 1.4 Redundancy (duplication) | Presence of overlapping solutions/measures and back-up systems; not cost-effective. Redundancy in the short-term to promote the best solutions in the long-term |
| **2. Learning Capacity** | |
| 2.1 Trust | Presence of authorities patterns that promote mutual respect and trust |
| 2.2 Single-loop learning | Ability of authorities patterns to learn from past experiences and improve their routines |
| 2.3 Double-loop learning | Evidence of changes in assumptions underlying authorities patterns (re-evaluates and reframes goals, values, etc.) |
| 2.4 Discuss doubts | Authorities openness towards uncertainties (deal with uncertainties) |
| 2.5 Institutional memory | Authorities provision of monitoring and evaluation processes of pathways/actions experiences |
| **3. Room for Autonomous Change** | |
| 3.1 Continuous access to information | Accessibility of data within authorities memory and early warning systems to individuals |
| 3.2 Act according to a plan | Increasing the ability of individuals to act by providing plans and scripts for action, especially in case of disasters |
| 3.3 Capacity to improvise | Increasing the capacity of individuals to self-organize and innovate; foster social capital |
| **4. Leadership** | |
| 4.1 Visionary | Room for long-term visions and reformist leaders |
| 4.2 Entrepreneurial | Room for leaders that stimulate actions and undertakings; leadership by example |
| 4.3 Collaborative | Room for leaders who encourage collaboration between different actors; adaptive co-management |
| **5. Resource** | |
| 5.1 Authority | Provision of accepted or legitimate forms of power; whether or not authorities rules/procedures are embedded in constitutional laws |
| 5.2 Human resources | Availability of expertise, knowledge and human labour |
| 5.3 Financial resources | Availability of financial resources to support measures and financial incentives |
| **6. Fair Governance** | |
| 6.1 Legitimacy | Whether there is public support for a specific authority |
| 6.2 Equity | Whether or not authorities rules/procedures are fair |
| 6.3 Responsiveness | Whether or not authorities patterns show response to society |
| 6.3 Accountability | Whether or not authorities patterns provide accountability procedures |

*2.2. Learning Loops*

Pahl-Wostl [23] argues that social learning is essential for developing and sustaining the capacity of different authorities, experts, interest groups, and the public to manage water resources effectively and translate goals into actions. Social learning occurs through interactions between actors within social networks [29]. Learning can be seen as a feedback process consisting of three iterative learning loops. Single-loop learning refers to improving the performance of established routines, i.e., are we doing what we do right? [23], for example, should the dyke height be increased by 10 or 20 cm? [30]. Double-loop learning reframes the goal, the problem and assumptions (e.g., about cause–effect relationships) within a value-normative framework, i.e., are we doing the right things? For example, what strategies might facilitate more effective future transboundary flood management? Or how should vulnerability to other climate change impacts be included in FRM? [30]. Learning outcomes include, for example, changes in the organization's knowledge base, new objectives, or new policy frames [31]. Triple-loop learning refers to a transformation of the structural context and factors that determine the frame of reference and reconsiders underlying values, beliefs, and world views [23], i.e., how do we decide what is right, for example, should resources be allocated toward protecting the existing built environment, or should these assets be relocated or abandoned once certain risk thresholds are crossed? [30]. Learning outcomes

include changes to defining principles—for example, underlying governance protocols and structures or new learning strategies [32].

The learning process can also be linked to pilot projects and their outcomes in relation to diversification of FRMS. Change in water governance is conceptualized as a stepwise approach from single to double and to triple-loop learning. It can be explained as a feedback loop between the expected outcomes within a specific governance context and considering the structural changes, which result in an iterative cycle (Figure 1). In this cycle, triple-loop learning is directly related to transformations, while single- and double-loop learning play an indirect role in these processes. Although social learning is key to learn from and support changes/transformations in water governance, empirical evidence is lacking [33]. More empirical research is needed that addresses the underlying triggers of double- and triple-loop learning processes and links them to broader governance mechanisms and structures [34].

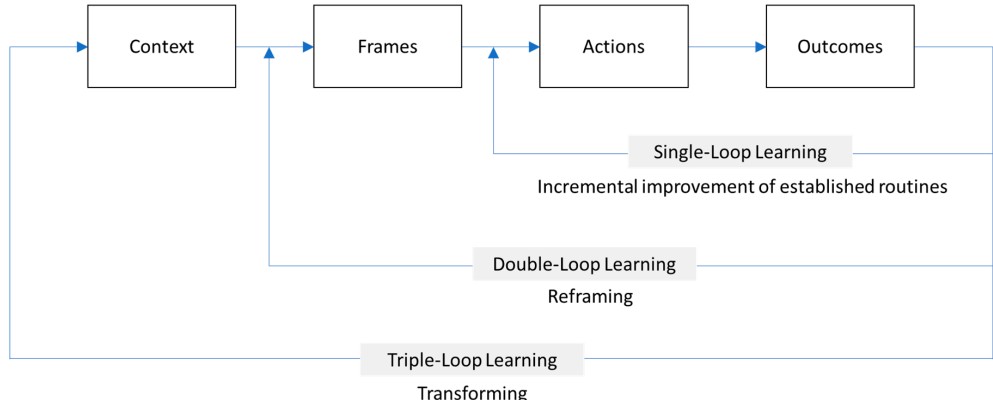

**Figure 1.** The concept of triple-loop learning applied to governance regimes [23].

## 2.3. Evaluating Adaptive Capacities and Learning Processes in Pilot Projects

In order to assess adaptive capacities and learning, the concepts of the ACW and the three loops of learning are combined in one analytical framework (Figure 2). Herewith, the framework aims to identify and analyze the adaptive capacities of governance networks and their importance for developing more diversified FRMS. Based on the Dynamic Adaptive Policy Pathways approach [35,36], six relevant steps for pilot projects were identified as part of adaptive planning processes. These steps are further divided into the situation before, during, and after the pilot projects. New diversified strategies, which include new actors, institutions, policies, and other aspects, require new knowledge, types of collaboration, and other capacities. Therefore, the goal is to understand how the concept of adaptive capacities supports the identification of requirements for diversified FRMS.

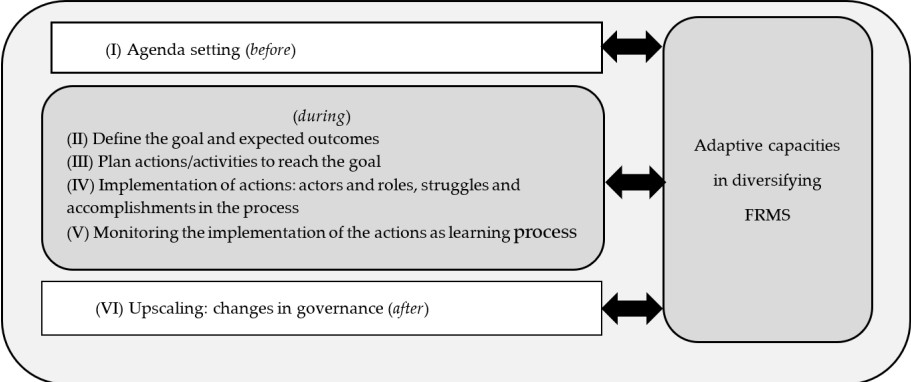

**Figure 2.** Framework for the analysis of adaptive capacities in pilot projects.

'Before' refers to the current FRMS as a result of the historical co-evolution of FRMS at the national and/or regional level within a specific governance setting [36]. It is important to have insight into this situation to understand the pilot outcomes within a specific governance context. Adaptive capacities that result in improvements of actions in the current strategy are considered as single-loop learning and have no direct effect on diversification of the current FRMS. 'During' refers to the implementation of actions during the pilot project and learnings from this implementation process. Employed adaptive capacities in this process can result both in single-loop learning and double-loop learning. The key question is whether developed adaptive capacities result in improvement and/or reframing of the FRMS. 'After' refers to outcomes of the pilot projects and upscaling of outcomes in order to change or transform the FRMS. Developed adaptive capacities lead to new frames (double-loop) or lock-in of the current strategy, with or without improvements of current actions (single-loop).

## 3. Research Questions

The main research question of this paper is: Which adaptive capacities support the learning process in pilot projects to achieve diversification of FRMS? The following three sub-questions are addressed:

(a)   Which adaptive capacities were determined to be missing before the pilot project?
(b)   Which adaptive capacities are employed during the pilot project?
(c)   Which adaptive capacities are developed as a result of the pilot project?

Missing capacities refer to capacities that are not present, or present but not applied in the governance network of the pilot project. Employed capacities refer to the capacities present and applied throughout the pilot project. Developed capacities refer to the capacities improved or emerged in the governance network as a result of the pilot project.

## 4. Materials and Methods

Data were collected as part of the Interreg FRAMES project [19] comprising 16 FRM pilot projects in five countries: Belgium, Denmark, England, Germany, and The Netherlands. Pilot projects varied substantially in terms of their content (e.g., risk analyses of critical infrastructure, implementation of nature-based solutions, increasing community resilience, see Table A1 in the Appendix A). For the purpose of comparison, two pilot projects with a similar focus were selected. In order to gain a broad understanding of the context and focus, both pilots were visited and explained by the involved stakeholders (Figure 1). During the pilot project, empirical data were collected through questionnaires, focus groups, and semi-structured interviews.

The questionnaires provided the initial and the expected state of flood resilience before and after the pilot project, respectively. Interviews were conducted with the pilot managers to get a more in-depth understanding of the specific lacking, employed, and developed capacities in the pilot projects. The transnational focus groups (TFG) were organized to gain insight into the most needed adaptive capacities for flood mitigation and flood preparedness actions in the pilot projects. Additionally, this data was complemented additional information from FRAMES project meetings and documents to make the finding more robust.

### 4.1. Pilot Projects

Two pilot projects were selected as case studies: the 'Alblasserwaard-Vijfheerenlanden' in The Netherlands and the 'Wesermarsch' in Germany. These two cases were selected from a total of 16 pilot projects in the FRAMES project (see Appendix A for an overview of pilots and selection). These pilot projects were selected because flood risk governance is comparable in both countries (Table 2). In both countries, defence/protection strategies are dominant but are also looking into more integrated strategies: Multi-layered safety [37] in the Netherlands and the LAWA approach in Germany [38]. This is part of a paradigm shift from a safety to risk-based approach [39–42]. Traditionally, responsibilities

in Dutch flood control are divided between the centralised Rijkswaterstaat and decentralised water boards [43]. Local actors are involved only when traditional defence approaches are not feasible anymore. In Germany, the federal states (Länder) have the main responsibility for all water issues and civil protection [44]. Actors from spatial planning play an important role in zoning plans or mitigation and flood risk management strategies in Germany [40]. However, the use of spatial planning instruments has increased in The Netherlands, as well as within flood risk management [43]. As part of developing the risk-based approach, both pilot projects aim to enhance the integration of mitigation and preparedness measures, including the development of evacuation strategies, raising risk awareness, and stimulating preparedness among citizens.

**Table 2.** Flood risk governance in countries of the selected pilot projects. Based on [43].

| Characteristics of Governance | The Netherlands | Germany |
| --- | --- | --- |
| Diversification and dominance | Low diversification, defence dominant | Moderate diversified, focus on defence |
| Multi-sector | Water sector dominant | Multi-sector involvement and integrated by spatial planning |
| Multi-actor | Public (state dominant) | Public (state and federal states) dominant |
| Multi-level | Both central and regional level | Central guidance and decentralization to federal state and local level |

### 4.2. Questionnaire

For each pilot project, a questionnaire (Appendix B) was completed by the pilot managers together with local relevant stakeholders. The pilot managers selected key stakeholders of their pilot project. In Alblasserwaard, the questionnaire was filled by four relevant stakeholders and in Wesermarsch, it was filled by seven local stakeholders (see Table A2, Appendix B). The actors filled out the questionnaire by organization prior to the start of the pilot project (October 2017) to identify the current (i.e., before the pilot) and the expected future (i.e., after the pilot project) level of diversification in the FRMS, and the perceived current and future level of flood resilience among authorities and communities. Closed questions were presented on rating scales 1 (not at all) to 10 (to a great extent). Open questions were asked to explain the indicated ratings. Finally, the answers to the open questions were aggregated, and scores to the closed questions were averaged.

Diversification. Using the disaster management cycle—pro-action (1), protection (2), mitigation (3), preparation (4), recovery (5)—stakeholders were asked 'To what extent is ( … ) a strong characteristic of the pilot area?' on a 1–10 rating scale. This question was asked for the current situation (i.e., before the pilot project) with respect to the main pillars of FRMS in The Netherlands and Germany; i.e., for pro-action (1) and protection (2). For the diversifying elements mitigation (3), preparation (4) and recovery (5), stakeholders answered this question both, for the current and the expected situation, after the pilot project. In total, stakeholders responded to eight items. This information was used to qualitatively estimate the ambitions of authorities in diversifying FRMS. Open questions were used to further tap into these ambitions for elements 3–5 by asking 'What will be done in the pilot with regard to ( … ) that improves the (a) 'physical resilience in the pilot area?', (b) 'capacities of local organisations/institutions in the pilot area?', and (c) 'capacities of local communities (citizens, businesses) in the pilot area?'

Resilience of authorities and communities. Stakeholders were asked to name the institutions and citizen groups that would be involved in and/or informed about the pilot project. Subsequently, stakeholders were asked to respond to the items: 'In general, to what extent is ( … ) embedded in policy and practice of these organisations, in your opinion?' and 'In general, to what extent is ( … ) embedded in the behaviour of these communities, in your opinion?' Both items were presented on a 1–10 rating scale. These items were presented separately for mitigation (3), preparation (4), and recovery (5), and in

the situation before the pilot project and the expected situation after the pilot project. Thus, overall stakeholders responded to twelve items. For each item, a written clarification was requested.

### 4.3. Interviews

Based on the theoretical framework, a comprehensive interview guideline (Appendix C) was developed and validated by the four involved knowledge institutes. The definitions of the adaptive capacities (Table 1) were integrated into an interview guideline through open questions to gain detailed insight into the opinions and arguments, but also to avoid social desirability bias. Interviewees were asked to reflect on past and current FRMS in the pilot project, any struggles encountered, the main accomplishments, the role of actors involved, and how these factors could contribute to mainstream the pilot project outcomes into the governance regime. All pilot managers were interviewed, and two interviews were used for the selected pilot projects (i.e., Alblasserwaard-Vijfheerenlanden, 28 February 2019; and Wesermarsch, 23 January 2019). The interviews were transcribed and checked by the pilot managers. Data analysis was performed using systematic coloured coding [45] to determine the adaptive capacities that were lacking, employed, and developed before, during, and after the pilot project. In order to facilitate data analysis, colours were assigned to each dimension of the ACW, and the criterions or adaptive capacities were numbered (see Table 1) to differentiate these in the interviews' transcripts and in the Results section. The results are presented in narratives to provide detailed storylines of the case studies [46]. The narratives of both pilot projects were reviewed by the pilot managers.

### 4.4. Transnational Focus Groups

Three transnational focus groups (TFG) were organized to gain insight into adaptive capacities before, during, and after the pilot projects and their relation to FRMS. This was done in parallel with the interviews of the pilot managers. Each TFG focused on a different FRM action (Table A3, Appendix D), that was typical for mitigation (via spatial planning), preparedness (integrating emergency response in FRM), and community resilience to address the actions by inhabitants in relation to mitigation, preparedness, and recovery. Each group was requested to select and discuss the five most relevant adaptive capacities (Table 1) needed for a specific action in an FRMS.

The three TFG were organized on 27 March 2019 in Oldenburg (Germany) and included 32 participants from five countries, representing (mainly regional) authorities with responsibilities in water management, spatial planning, crisis management, and community resilience. In order to facilitate transnational learning, each TFG consisted of participants from the 16 pilot projects and countries (Appendix D). Each group was moderated by an author of this paper.

### 4.5. Additional Information

In addition to the interviews held with pilot managers, presentations by other stakeholders of the pilot projects were attended during visits to the pilot projects (Alblasserwaard-Vijfheerenlanden, 22 February 2019; Wesermarsch, 28–29 March 2019). The presentations and discussions provided more background and insights into the role that these actors played in the pilot projects and FRMS.

## 5. Results

### 5.1. Diversification of FRMS in Two Pilot Projects

Both pilot projects are located in low lying areas that are exposed to flood risks. Results from both questionnaires (Table 3) confirm that current FRMS lean on flood protection by dykes. Protection levels are regarded as 'high', but further enforcements are required to meet the legal flood protection standards. Flood preparedness can be regarded as the second pillar in the current situation as it has been embedded already in current policy and practice of crisis management authorities. However, both pilot projects aimed to make small improvements in flood preparedness. Mitigation is less developed,

and both pilot projects aimed for improvement. Recovery is least developed and is not particularly focused on in the pilot projects.

To achieve the improvements in mitigation and preparedness among authorities, the Dutch pilot project expected that stronger involvement of local authorities would increase awareness and lead to improvements in spatial planning (mitigation) and better evacuation plans (preparedness). Currently, stakeholders rely strongly on protection measures, increasing their knowledge and awareness will take time. Likewise, embedding flood risk mitigation and preparedness in current policies and practices also requires long term planning. Similarly, the German pilot project focused on increased involvement of actors and cooperation aiming to improve the incorporation of flood risk in regional planning (mitigation) and to improve contingency planning by reviewing plans collectively (preparedness).

In order to improve mitigation, especially preparedness among communities, both pilot projects focused on providing better information about the warning systems and increasing their knowledge and awareness.

**Table 3.** Perceived diversification of risk management strategies (FRMS) before and expected after the pilot projects. Scores on scale 1–10 (* left number refers to before and right number after the pilot project).

| Pilot | Aspects of Embeddedness of FRMS | Pro-Action (1) | Protection (2) | Mitigation (3) | Preparation (4) | Recovery (5) |
|---|---|---|---|---|---|---|
| Alblasserwaard NL | Strong characteristic of area | 5 | 10 | 4–5 * | 7–6 | 2–2 |
| | Embedded in policy and practice of authorities | | | 5–4 | 8–7 | 3–2 |
| | Embedded in the behaviour of communities | | | 5–4 | 7–6 | 2–2 |
| Wesermarsch GE | Strong characteristic of area | 3 | 8 | 5–3 | 8–7 | 1–1 |
| | Embedded in policy and practice of authorities | | | 6–5 | 7–6 | 1–1 |
| | Embedded in the behaviour of communities | | | 4–3 | 7–6 | 1–1 |

*5.2. Lacking, Employed, and Developed Adaptive Capacities in Both Pilot Projects*

Table 4 summarizes the main characteristics of both pilot projects. Neither of the pilot projects had upscaling as its objective, but through interviews, several capacities needed for upscaling were identified that were not present.

**Table 4.** Main characteristics of both pilots.

| Characteristic | Alblasserwaard-Vijfheerenlanden | Wesermarsch |
|---|---|---|
| Goal of the pilot | Mitigation: combine evacuation with spatial planning | Mitigation: improve spatial database use for crisis management |
| | Preparedness: improve emergency management (risk maps, evacuation routes, etc.), investigate social capital for emergency evacuation, improve collaboration between water and crisis management authorities | Preparedness: improve emergency planning, provide access to citizens about self-preparedness in case of flooding, improve collaboration between water and crisis management authorities |
| | Recovery: no direct actions but increase in flood risk awareness will decrease recovery actions | Recovery: no direct actions but increase in flood risk awareness will decrease recovery actions |
| Lead Organization | Province of Zuid Holland | Jade University of Applied Sciences |
| Main actors involved | Ministry of defence | NLWKN: State Agency on water management, coastal protection and nature conservation |
| | Rijkswaterstaat | Aid organisations: German Red Cross, Technisches Hilfswerk, Federal Armed Forces, Police, German Lifeguard Association |

**Table 4.** *Cont.*

| Characteristic | Alblasserwaard-Vijfheerenlanden | Wesermarsch |
|---|---|---|
| | Safety Region | Drinking water company (Oldenburgisch-Ostfriesischer Wasserverband) |
| | Water board | Regional power supplier (Energieversorger Weser) |
| | 3 Local municipalities | Wesermarsch County |
| | Drinking water company | Municipality of Butjadingen |
| | Local businesses | Local groups: farmers' association and veterinary group |
| | | 6 water boards and 2 dyke boards |
| | | LAVES: State Agency for food security |
| Timeline of the project | 2016–2019 | 2016–2019 |

### 5.2.1. Case Study 1: Alblasserwaard-Vijfheerenlanden Pilot Project

The goal of this pilot project was to achieve more diversified FRMS by linking spatial planning and emergency management. The main characteristics of the pilot project, the pilot goal, the leading organization, the actors involved, and the duration of the project can be found in Table 4. The specific actions of this pilot project were to ensure that dyke roads are suitable for emergency vehicles, to build social capacity for evacuation involving local groups, and to improve evacuation management plans in case of flooding.

The pilot project faced a number of difficulties during set-up, that were identified as 'lacking capacities before the pilot project' (Table 5, indicated with - in column 'before'). The pilot manager determined that there was a lack of variety in FRM solutions (1.3) in the Alblasserwaard area because the FRMS rely mainly on protection. The second difficulty encountered was that local mayors had a lack of information about the diversification of FRMS. They thought that FRM beyond hard infrastructure was not possible or limited (e.g., existing evacuation plans are not clear). Moreover, there was low social capital (3.3) because there were not enough volunteers to improve preparedness and heavy reliance on governmental flood protection. The third issue was a lack of collaboration (4.3) between crisis and water management authorities. For example, prior to this pilot project, the safety region, water board, and local municipalities did not collaborate on integrated flood risk management projects (combining evacuation and spatial development). The fourth difficulty was limited human resources. Local municipalities have limited staff (5.2), and sometimes, one person is responsible not only for FRM but many other subjects.

In order to define the goal of the pilot project and implement the planned activities, multiple capacities were employed during the pilot project (Table 5, indicated with + in column 'during'). The implementation of diversified FRMS required the involvement of multiple actors (1.2) from a variety of levels and sectors (the safety region, the province, the water board, and municipalities) at an early stage of the pilot project. They met regularly to discuss problems and needs (1.1) related to the current FRMS and together came up with a diversity of solutions (1.3). For example, one of the problems was that emergency vehicles might not be able to use the dyke roads during evacuation because the water board needs it for equipment or because of the risk dyke failure due to instability under extreme conditions. Based on the discussions among the actors involved, the agreement was to align their needs in case of future dyke reinforcements integrating protection, infrastructure planning, and emergency response. The actors involved gathered in regular project meetings to define the (long term) goals (4.1) and specific objectives of the pilot project. The pilot manager highlighted that collaboration (4.3) was positive during the pilot project, and early collaboration started with existing initiatives in the area. Problems, needs, solutions, and next steps were discussed (2.4) with the actors. As a result, learning from the past and the current actions in FRMS (2.2) during the pilot project is an essential capacity. Moreover, pilot projects offer a higher availability of resources when human capacity (5.2) and finances (5.3) are combined to experiment with innovative solutions in FRM. For example, the safety region had the human capacity for modelling the evacuation routes while the province had the financial resources for it.

**Table 5.** Adaptive capacities before, during, and after the pilot projects and their perceived relevance for FRMS.

| Dimensions of the ACW | Lacking, Employed, and Developed Capacities | | | | | | Perceived Relevance of Capacities | | |
|---|---|---|---|---|---|---|---|---|---|
| | Before | | During | | After | | Mitigation | Preparedness | Com. Resilience |
| Adaptive Capacities | Albl. | Weser. | Albl. | Weser. | Albl. | Weser. | | | |
| **1. Variety** | | | | | | | | | |
| 1.1 Variety of problems frames | | | + | + | | | | | |
| 1.2 Multi-actor, -level, -sector | | | + | + | | | • | • | |
| 1.3 Diversity of solutions | - | - | + | + | # | # | • | | |
| 1.4 Redundancy of solutions | | | | | | | | | |
| **2. Learning Capacity** | | | | | | | | | |
| 2.1 Trust | | | | + | | # | | | • |
| 2.2 Single-loop learning | | | + | + | # | # | | | |
| 2.3 Double-loop learning | | | + | + | | | | | |
| 2.4 Discuss doubts | | | + | + | | | | | |
| 2.5 Institutional memory | | | | | # | # | | | |
| **3. Room for Autonomous Change** | | | | | | | | | |
| 3.1 Continuous access to information | - | - | + | + | # | # | • | • | • |
| 3.2 Act according to a plan | | | | | | | | | |
| 3.3 Capacity to improvise | - | - | + | + | # | # | | | • |
| **4. Type of Leadership** | | | | | | | | | |
| 4.1 Visionary | | | + | + | - | | • | | |
| 4.2 Entrepreneurial | | - | | + | - | - | | | |
| 4.3 Collaborative | - | - | + | + | # | # | | • | • |
| **5. Resources** | | | | | | | | | |
| 5.1 Authority | | | | | | | • | | |
| 5.2 Human resources | - | - | + | | - | - | | • | • |
| 5.3 Financial resources | | | + | | | - | • | | |
| **6. Governance** | | | | | | | | | |
| 6.1 Legitimacy | | | | | | | | | |
| 6.2 Equity | | | | | | | | • | |
| 6.3 Responsiveness | | | | | | | | | |
| 6.4 Accountability | | | | | | | | | |

Albl. = Alblasserwaard pilot project, Weser. = Wesermarschpilot project. - refers to lacking capacities; + refers to employed capacities; # refers to developed capacities; • refers to 5 most important adaptive capacities selected by the transnational focus groups (TFG). The grey colour reflects the lacking capacities before but developed as a result of the pilot projects, which were also emphasized as important capacities by the TFG.

Based on the pilot project outcomes, several capacities were developed (Table 5, indicated with **#** in the column 'after'). The first and the most important developed capacity was that more innovative solutions were applied in the pilot project to diversify the FRMS (1.3) by combining evacuation with spatial development. The second capacity that was developed as a result of the pilot project is the collaboration between actors (4.3) with respect to crisis management improved. For example, actors (from the Safety Region, the water board, and municipalities) who did not collaborate on this subject prior to this pilot project, currently they understand the significance of linking evacuation and spatial planning in relation to FRM. A relevant outcome of this pilot project is providing more detailed information (3.1) about the applicability of diversified FRMS in the area and making it accessible for everyone. For example, when this information was provided to the municipalities, they expressed more interest to participate in the project. Thus, sharing information resulted in capacity building (3.3) and awareness of the role of citizens and local authorities in evacuation management. Another developed capacity is the learning from pilot projects when actors share knowledge and learn from each other. For instance, during a meeting with local entrepreneurs, it became clear that several local entrepreneurs had already taken actions to protect themselves against floods in the current FRMS (2.2). Moreover, at the end of the pilot project, the lead authority provided a final report that included pilot project results and policy recommendations. This was an evidence-based document (2.5) that can be used by other actors to replicate the pilot project.

Finally, two capacities were identified as 'lacking capacities for upscaling of the pilot project' (Table 5, indicated with - in column 'after'). On the one hand, the pilot manager pinpointed that visionary leadership (4.1) is required in the process of diversification of FRMS. However, this can be hampered, for example, by a change of government representatives every four years. Likewise, it is not clear who takes the lead (4.2) to adapt or change current policies based on the pilot project outcomes. On the other hand, human resources (5.2) are lacking because generally, governments do not have enough human capacity to cooperate in projects related to the diversification of FRMS.

5.2.2. Case Study 2: Wesermarsch Pilot Project

The goal of the Wesermarsch pilot project was to develop a more balanced FRMS, taking actions in spatial planning (mitigation) and emergency management (preparedness). The main characteristics of the pilot project, the goal of the pilot project, the leading authority, the actors involved, and the duration of the project are described in Table 4. The actions of the pilot project are to develop informative products (brochure for both individual farmers preparedness), to improve the current database with flood risk maps, to develop an app to organize volunteers during a disaster, and to organize a 'flood risk awareness day' and a 'flood partnership event'.

'Lacking capacities before the pilot project' were identified by using data from a previous EU project [47], and the outcomes of interviews conducted by Jade University (Table 5, indicated with - in column 'before'). Traditionally, the main focus in the Wesermarsch is on flood protection through hard infrastructure and far less on flood mitigation through spatial planning, and preparation and recovery from a flood event. Therefore, it was determined that there were insufficient varieties of solutions (1.3) in FRMS. Moreover, when the EU previous project ended [47], the communication and further actions between the actors involved (water management actors) also stopped. However, generally in Germany, there is a shortage of collaboration (4.3) among authorities responsible for diverse aspects of the current FRMS. Likewise, there are no local groups who take the lead (4.2) to prepare volunteers in case of an evacuation. Furthermore, it was also identified that citizens had limited access to information (3.1) about flood risk preparedness and thus, they had a low social capital (3.3) in order to prepare themselves in case of flooding. An additionally lacking capacity was an unsuitable procedure (5.2) for the use of spatial databases (e.g., elevation maps, evacuation routes, etc.) to improve decision making for spatial planning at a local level.

During the pilot project, different capacities were employed (Table 5, indicated with + in the column 'after'). First, the pilot manager involved multiples actors (1.2) in the pilot process (local and regional actors from disaster, water, flood risk management, and aid organizations, see Table 4). The actors met and discussed the diversification of FRM problems (1.1) to provide diverse solutions (1.3) for these problems. For example, flood risk maps used for evacuation exercises did not consider the topography behind the dyke, and no local group existed to organize volunteers in case of a disaster intervention. Secondly, leadership was promoted through the implementation process of the pilot project. Regional fora were organized (4.3) where key actors discussed problems and solutions. Afterwards, a priority list of activities with a long term vision (4.1) was made. Six activities were selected, and voluntarily actors took the lead (4.2) to implement them. Thirdly, the previous project [47] in the area was utilized as a learning example (2.2) to find out what did or did not work in the past and why. Likewise, the actors' discussions about undefined challenges (2.4) also contributed to the learning process.

As a result of the pilot project, multiple capacities were developed (Table 5, indicated with # in the column 'after'). The pilot outcomes resulted in a diversity of solutions (1.3) for FRM in the Wesermarsch area. For example, increased self-preparedness of local citizens and an improved spatial database for spatial emergency management planning. The development of informative products, regional fora to exchange information, and experiences resulted in diversified FRMS. Therefore, making information more accessible (3.1) for citizens leads to improved flood risk awareness and social capital (3.3). On top of that, the collaboration (4.3) between local and regional actors from disaster and water management

authorities was enhanced. As a result of the collaborative capacity, the pilot manager observed an increase of trust (2.1) and improved communication about diverse FRMS. In addition, the pilot results showed improvements in preparedness (2.2) due to the actions taken. For example, farmers and other inhabitants are better informed about FRM and evacuation planning; flood risk maps at the local level are improved, and these will be used by the German state agency to support emergency planning decisions. The pilot project actions and outcomes are publicly accessible on paper and online (database, reports, flyers). This evidence base (2.5) can be used to learn from this pilot project and replicate it.

The interview with the pilot project manager revealed two capacities that were identified as 'lacking capacities for upscaling of the pilot project' (Table 5, indicated with - in column 'before'). The first one is the lack of entrepreneurial leadership because the leading authority of this pilot project has no formal responsibilities in FRM. Moreover, none of the other organizations involved have shown the initiative to take on responsibilities (4.2) regarding upscaling. The second lacking capacity is insufficient human (5.2) and financial (5.3) resources to update the developed pilot project outcomes when needed by the responsible authorities.

*5.3. Perceived Importance of Adaptive Capacities in Diversifying FRMS*

Three TFG focused on a different type of strategy to diversify the current FRMS, and each TFG selected five adaptive capacities that were evaluated as having the highest relevance for implementing these strategies. TFG1 focused on mitigation, TFG2 on preparedness, and TFG3 on emergency response. Table 5 presents an overview of the capacities that were linked to specific strategies.

The TFG1 (mitigation) focused on spatial zoning in flood prone areas to mitigate flood risk. Visionary long-term planning (4.1) was highlighted as having a high relevance. All pilot projects in FRAMES face long term challenges considering flood risk and other climate related issues. The TFG stressed that sharing information (3.1) about future projects between spatial planners and water managers is needed. Stakeholders perceived that the attention for long term goals is hampered by short term gain, and often, economic benefits on the short term prevail. Authority by law via procedures (5.1) and financial resources (5.3) were seen as important resources to arrange mitigation measures. In addition, they stressed the need for a variety of stakeholders (1.2) who provide a diversity of solutions (1.3).

TFG2 (preparedness) looked into flood preparedness by integrating emergency response planning in flood risk management. The group indicated that available human resources (5.2) are key adaptive capacities. There is a need for specialized staff that has the expertise to perform an impact assessment of flooding to aid preparedness and emergency response. Moreover, equity (6.2) is considered important in flood preparedness. Different societal groups have different needs in case of emergency, for example, less self-reliant (elderly, disabled) people depend on the assistance of authorities. This focus group also highlighted that emergency response could be supported by spatial planning measures, for instance, by building public shelters and safe evacuation roads. A diversity of actors (1.2) responsible for spatial planning and crisis management should collaborate (4.3) and share information (3.1) about their current policies and projects.

TFG3 (community resilience) focused in more detail on empowering communities to take action for local flood mitigation and response measures. The TFG emphasized it is important to encourage collaboration (4.3) and build trust (2.1) between responsible authorities and communities. Stronger collaboration capacity contributes to higher availability volunteers (5.2) in emergency preparedness. The capacity to provide relevant stakeholders with access to information (3.1) about mitigation and preparedness measures for communities appears to be an important issue. Raising awareness about the measures that communities can take by themselves fosters social capital (3.3).

## 6. Discussion

This paper aimed to identify the adaptive capacities that support the learning process in pilot projects to achieve a diversification of FRMS. Through questionnaires, focus groups, and interviews, we

focused on the adaptive capacities that were lacking, employed and developed before, during, and after two pilot projects in the Netherlands and Germany. The results showed that in both pilot projects, the current FRMS leans traditionally on flood protection, with flood preparedness as a secondary pillar. Flood mitigation, but especially recovery strategies, were hardly present. The learning process in both pilot projects focused on strengthening flood preparedness and mitigation, for instance, by involving new stakeholders, sharing knowledge, reviewing contingency plans, and by providing information to citizens to increase knowledge and awareness.

We found three adaptive capacities that were stressed as important for developing more diversified FRMS and that were also lacking before but had developed as a result of the pilot projects. First, a greater 'diversity of solutions' was regarded as important, especially for developing flood mitigation strategies but not for flood preparedness and community resilience. The reason for this is that flood mitigation is currently underdeveloped and requires a balanced mix of cost-effective spatial planning actions. Finding cost-effective spatial planning measures is difficult since flood defences act as a 'front door' which make any investments in the area behind this front door redundant [48,49]. In the current frame of FRM, clear added benefits first need to be identified to gain political support for these investments, which seems difficult. Increased opportunities for integrating spatial planning in FRMS, therefore, requires reframing of current FRM policy and practice. In the Netherlands, such reframing has partly taken place with the adoption of the multi-layered safety concept (protection, spatial planning, crisis management) in FRM policy [50]. However, because the basic question 'are we doing what we do right?' (single-loop learning) has not changed to 'are we doing the right things?' (double-loop learning). Pilot projects have not succeeded in putting more diversified FRMS into practice [16,48]. For instance, there is currently little urgency to consider the meaning of a wider set of challenges originating from long term processes such as soil subsidence and sea level rise [51]. Since such challenges are not yet fully incorporated into the current FRMS, also at the level of pilot projects they are hardly considered. Reframing the problems and goals of FRM, therefore, requires the inclusion of a 'variety of perspectives over problems/needs' beyond FRM. The processes that steer this type of fundamental reframing require learning and governance capacities, which were not prioritized as important capacities in the pilot projects due to the incremental improvements (single-loop learning) that were aimed for.

Second, to create room for autonomous change, authorities and communities require greater access to information. Although this may sound obvious, the challenge is in making the right information accessible for the different stakeholder groups. Information preferences may differ substantially between actors in terms of information type, detail, and ways of receiving information (e.g., channel, format), for instance, a step-by-step checklist for farmers to prepare themselves and their livestock (Wesermarsch) and how entrepreneurs can protect their businesses (Alblasserwaard-Vijfheerenlanden) in case of flooding. Making the information on emergency planning available for the actors resulted in enhanced mutual understanding of interests, actions, and information needs.

Third, type of leadership was regarded as an important antecedent of diversifying FRMS. Actors agreed that collaborative leadership, encouraging the collaboration among actors, is currently needed to further develop preparedness and community resilience. The literature supports that collaborative networks are essential for performing adaptive management [52,53]. Alignment across sectoral boundaries is key in governance arrangements for adapting to climate change [54], which is also observed in both cases. Boundary spanning interactions, including cherishing boundaries for clear allocation of responsibilities [55], is required for collective action in diversifying FRMS. Since mitigation strategies are underdeveloped and complex, visionary leadership seemed more important for developing cost-effective spatial planning strategies. These strategies were employed during the pilot projects; however, the capacity to develop 'long term goals and strategies' did not result from the pilot projects. This aligns with their focus on incremental improvements.

Furthermore, the adaptive capacity dimension 'resources' received some importance ratings for diversifying FRMS. Law, procedures, and policy development, as well as financial resources,

were regarded as important for developing mitigation strategies. Human resources, such as knowledge, expertise, and availability of volunteers, were regarded as important for developing preparedness and community resilience. However, during the pilot projects, none of these capacities were developed, which can be seen as a risk for further uptake of the outcomes of the pilot projects [16].

Additionally, two capacity dimensions hardly received importance ratings. First, within the 'learning capacity' dimension only, trust was regarded of some importance for building community resilience. This is remarkable because the interviews with pilot managers did show that learning capacities were employed and improved as a result of the pilot projects. The reason that learning was not identified as an important capacity may be explained because, as stated previously, the pilot projects rather focused on single-loop learning (incremental improvements of established routines) instead of double-loop learning (reframing of the FRMS) or triple-loop learning (transformation of FRMS). This is also supported by the pilot managers ambitions of diversifying the FRMS through the pilot projects (i.e., small incremental improvements were expected in mitigation and preparedness). This aligns with planning literature, which emphasizes that planning practices are more adaptive (adjust to changing circumstances) and incremental (gradual changes) than often assumed by scholars proposing 'new' planning approaches [56,57]. The interviews in our study showed that stakeholders learning capacities improved as a result of the pilot projects. Second, none of the capacities related to the dimension 'governance' emerged from the interviews, and governance was hardly regarded as an important antecedent for mitigation, preparedness, and community resilience. Governance may have gained little attention because most of the governance dimensions are already well institutionalized in the current arrangements of FRMS and, therefore, little action is needed to improve governance capacities in the current FRMS. Again, because current FRMS are well developed and institutionalized, improving weak links in the current governance regime is challenging. For instance, in a review of Dutch water governance, the OECD has pointed to a lack of awareness and preparedness among citizens and the large distance between water institutions and the general public [58]. Since society has a high level of trust in FRM, there is little urgency to bridge this gap, neither by the institutions nor by members of the general public. As a result, governance capacities become a passive part of FRM and fall short in gaining public support, responding to (implicit) information needs in society and taking responsibility for providing information about preparedness and response strategies. The lack of importance ratings for governance shows that there was little awareness for this underlying mechanism, likely because the pilot projects did not fully enter the process of double or triple-loop learning. In addition to more urgency [51], more research is also needed about the role of pilot projects in transitions processes. The pilot projects studied in this paper appeared to be examples of incremental change in the diversification of FRM. Considering wicked problems like sea level rise, these pilot projects can be considered as small wins [59]. Taking the contextual dynamics of experiments into account, the studied pilot project matched best with a seedbed lens [60]. The protective environment of the Interreg project FRAMES creates an environment to develop new FRM actions and learn from these. The propelling mechanism framework by Termeer and Dewulf [59] is relevant for future research to evaluate the transformation potential of various small wins. Recent expectations about sea level rise [51] can result in a change of the contextual dynamics of flood resilience pilot projects, in which battleground experiments [60] could become more relevant.

The Governance Capacity Framework (GCF) developed by Koop et al. [61] and applied by Brockhoff et al. [62] has many similarities with the ACW framework applied in this paper (see [62] for a comparison of both frameworks). The main difference between the GCF paper [62] and this paper is in the application aim. We have applied the ACW to assess the capacity development of practitioners in pilot projects, while the GCF aims to assess the governance capacity of society to solve specific challenges [61,62]. This results in differences in the applied methodology. In this paper, we have combined the ACW with Triple-loop Learning and applied this as a qualitative approach without scoring the adaptive capacities. With case narratives and focus groups, we have aimed to gain insight into the development of adaptive capacities by pilot projects over time and identification of key

capacities for diversified FRM. In the GCF approach, Brockhoff et al. [62] scored the current governance capacity of cities and prescribed what steps involved practitioners need to take. The combination of both methods can be complementary in future research by combining scores to assess the current status and development of governance capacity. The indicator scoring of capacities is valuable for comparing scores of multiple cases. The qualitative approach, as applied in this paper, provides a more in-depth insight into the development of adaptive capacities in the context of specific actions for flood resilience.

## 7. Conclusions

In this study, an analytical framework was proposed combining the ACW and Triple-loop Learning to assess capacity development in pilot projects. The combination of these two approaches is a unique outcome of this paper. It acknowledges the development of adaptive capacities as a result of pilot projects and enables to link this with three types of learning. The findings contribute to theories about niche–regime interactions [20] and policy transfer via pilot projects [63]. The ACW within the framework was used as a qualitative approach without scoring the adaptive capacities [22]. The narratives allowed to pinpoint the development and interdependencies between adaptive capacities over time [22] in the phases (before, during, and after) of the pilot projects. Therefore, this analytical framework is practical to assess the development of capacities of stakeholders in pilot projects that aim to diversify FRMS. Likewise, it also identifies lacking capacities that are needed to ensure successful pilots and uptake in policy.

Since the proposed framework is the product of ongoing research, much room for improvement exists. Here, we mention a few avenues needing improvement. First, the framework misses clear guidance to evaluate the success of pilots and upscaling of pilots in the policy regime. By assessing pilot goals and outcomes more explicitly, the evaluation process can be improved. In particular, we regard the 'pilot paradox' [16] as a valuable approach because it defines the conditions underlying this process. Interestingly, the pilot paradox argues that the same conditions that make pilots successful often hamper their uptake in policy. Propelling mechanisms can help to assess the transformation potential of pilot projects as small wins in the domain of climate change [59]. Second, the methods of identifying adaptive capacities can be improved by incorporating multi-item rating scales to increase the reliability of measurements. This is a common approach in questionnaire research and provides a strong asset for further validation of the framework in future studies. Lastly, in order to get a better understanding of the capacity development in the pilot projects, it is necessary to consider different actors' perspectives. Therefore, it is recommended to conduct interviews with a variety of stakeholders of the pilot projects. This can strengthen the framework and application in future studies.

To conclude, this study has shown that the analytical framework is valuable for assessing pilot projects and learning about capacity development in their transition towards diversified FRMS. This methodology is a unique outcome of the FRAMES project, and its applicability to this study contributes to the existing literature about diversification of FRMS.

**Author Contributions:** Conceptualization: F.S.C., J.-M.B., M.B. and T.T.; data curation: F.S.C., J.-M.B., M.B. and T.T.; formal analysis: F.S.C., J.-M.B., M.B. and T.T.; investigation F.S.C., J.-M.B., M.B. and T.T.; methodology: F.S.C., J.-M.B., M.B. and T.T.; supervision: F.S.C., J.-M.B., M.B. and T.T.; writing—original draft preparation: F.S.C., J.-M.B., M.B. and T.T.; writing—review and editing: F.S.C., J.-M.B., M.B. and T.T.

**Funding:** This research was funded as part of FRAMES [19], an Interreg project supported by the North Sea Programme of the European Regional Development Fund of the European Union.

**Acknowledgments:** We want to give thanks to the knowledge institutes (Ghent University, University of Oldenburg, Jade University of Applied Sciences) for their contribution in developing the interview guideline. Moreover, special thanks to the pilot managers to provide valuable input and reviews from their pilots. Additionally, we want to thank all FRAMES project partners for their participation and input through the questionnaires and focus groups discussions.

**Conflicts of Interest:** The authors declare no conflict of interest.

## Appendix A  Selection of Cases, Based on Intended FRM Actions in Pilot Projects

Table A1 below provides an overview of the pilot projects in FRAMES and their intended actions considering diversification of flood risk management strategies (FRMS). All pilot projects, actions, and implementation processes are more detailed described on www.frameswiki.eu.

The selection of the Alblasserwaard and the Wesermarsch pilots was based, firstly, on that the idea that traditional flood management relies mainly on flood defence with hard infrastructure. Selection of the two cases for this paper was based on the intended integration of Mitigation (via spatial planning) and Preparedness (awareness raising and evacuation planning) (see Table A1). In addition to the two selected cases, the Sloegebied pilot project also intended to focus on the same aspects, but this pilot project just started when we started with the analysis for this paper.

**Table A1.** Flood risk management actions implemented in FRAMES all pilot projects.

| Pilot Projects | Flood Risk Management Actions | | | | | | |
| --- | --- | --- | --- | --- | --- | --- | --- |
| | Mitigation via Spatial Planning | Preparation (Awareness, Evacuation Planning) | Natural Flood Man. | Critical Infra-Structure | Recovery | Adaptive Planning | Community Resilience |
| 1. Alblasserwaard (NL) | | | | | | | |
| 2. Flood proof electricity grid Zeeland (NL) | | | | | | | |
| 3. Reimerswaal (NL) | | | | | | | |
| 4. Sloegebied (NL) | | | | | | | |
| 5. Wesermarsch (DE) | | | | | | | |
| 6. Ninove (BE) | | | | | | | |
| 7. Denderleeuw (BE) | | | | | | | |
| 8. Geraardsbergen (BE) | | | | | | | |
| 9. Kent (UK) | | | | | | | |
| 10. Medway (UK) | | | | | | | |
| 11. Southwell (UK) | | | | | | | |
| 12. Lustrum Beck (UK) | | | | | | | |
| 13. Butt Green Shield (UK) | | | | | | | |
| 14. Roskilde (DK) | | | | | | | |
| 15. Assens (DK) | | | | | | | |
| 16. Vejle (DK) | | | | | | | |

Light grey colour represents the diversified FRM actions implemented in each pilot project while dark grey colour shows the FRM actions' resemblance between the selected pilot projects for this study.

## Appendix B  Monitoring Survey (FRAMES)

Baseline measurement
Pilot: … … … … … … … .
Important:

We kindly ask the pilot manager to complete this questionnaire in consultation with relevant experts/stakeholders in the pilot/region. The pilot manager can send this questionnaire and ask these experts/stakeholders to complete (certain) questions, or ask them to review answers. Per pilot we would like to have 1 questionnaire returned.

Who contributed to completing this questionnaire? Table A2 provides the organisations and the function of the stakeholders who filled the questionnaire for both pilots, Alblasserwaard and Wesermarsch.

**Table A2.** Stakeholders who filled the questionnaires for Alblasserwaard (left) and Wesermarsch (right) pilot projects.

| | Organization | Function | | Function | Function |
|---|---|---|---|---|---|
| 1 | Province of Zuid-Holland | Pilot manager/Policy advisor | 1 | Jade University | Pilot manager/full professor |
| 2 | Rijkswaterstaat | Policy advisor | 2 | Jade University | Researcher |
| 3 | Regio Alblasserwaard en Vijfheerenlanden | Policy advisor | 3 | Landkreis Wesermarsch Wesermarsch | Head of Crisis Management at the County |
| 4 | | Policy advisor | 4 | Wesermarsch, Küste and Raum | Consultant in Coastal Management |
| 5 | Noordelijke Drechtsteden | | 5 | Drinking Water Company OOWV | Spatial Planning Manager |
| 6 | | | 6 | Drinking Water Company OOWV | Asset, Strategic Planning Manager |
| 7 | | | 7 | Drinking Water Company OOWV | Consultant Innovation Networks |
| 8 | | | 8 | | |
| 9 | | | 9 | | |
| 10 | | | 10 | | |

Short Explanation

FRAMES [19] is about improving flood resilience by taking different types of actions. A common and well-known typology in flood risk management is the disaster management cycle, as shown in the diagram below. This survey first focuses on the flood resilience of areas by considering the five elements of the Flood Risk Management cycle (Figure A1): pro-action (1), protection (2), mitigation (3), preparation (4) and recovery (5). Hereafter, a few questions will be asked with regard to the flood resilience of communities and authorities.

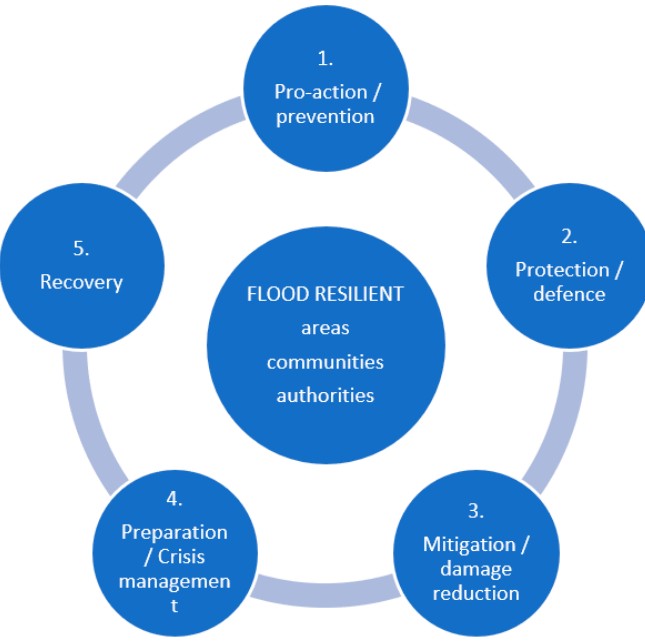

**Figure A1.** Flood Risk Management cycle.

*Appendix B.1 Diversification of FRMS:*

Appendix B.1.1 Pro-Action/Prevention

Negative consequences of flooding can be avoided by pro-active spatial planning or land use policies ('keeping people away from water'), aimed at building only outside areas that are prone to flooding.

To what extent is pro-action/prevention currently a strong characteristic of the pilot area?

| Not at All | | | | | | | | | To a Great Extent |
| --- | --- | --- | --- | --- | --- | --- | --- | --- | --- |
| 1 | 2 | 3 | 4 | 5 | 6 | 7 | 8 | 9 | 10 |

Please shortly explain your answer:

Appendix B.1.2 Flood Protection/Defence

Keeping water away from people by (combinations of) hard infrastructural works (dykes, dams, etc.) or softer (nature based) solutions (dunes, retention in nature areas, etc.).

To what extent is flood protection/defence currently a strong characteristic of the pilot area?

| Not at All | | | | | | | | | To a Great Extent |
| --- | --- | --- | --- | --- | --- | --- | --- | --- | --- |
| 1 | 2 | 3 | 4 | 5 | 6 | 7 | 8 | 9 | 10 |

Please shortly explain your answer:

Appendix B.1.3 Flood Risk Mitigation

Consequences of floods can be mitigated by a smart design of the flood-prone area including spatial orders, constructing flood compartments, or (regulations for) flood-proof building.

To what extent is flood risk mitigation a strong characteristic of the pilot area?

| | Not at all | | | | | | | | | To a great extent |
| --- | --- | --- | --- | --- | --- | --- | --- | --- | --- | --- |
| Currently | 1 | 2 | 3 | 4 | 5 | 6 | 7 | 8 | 9 | 10 |
| Expectation after FRAMES | 1 | 2 | 3 | 4 | 5 | 6 | 7 | 8 | 9 | 10 |

What will be done in FRAMES with regard to flood risk mitigation, that improves the

1.	physical resilience in the pilot area?
2.	capacities of local organisations/institutions in the pilot area?
3.	capacities of local communities (citizens, businesses) in the pilot area?

Appendix B.1.4 Flood Preparation

Consequences of floods can be mitigated by preparing for adequate response to a flood event. Measures include flood warning systems, disaster management and evacuation/rescue plans, and managing a flood when it occurs by taking last call emergency measures (e.g., sand bags).

To what extent is flood preparation a strong characteristic of the pilot area?

| | Not at all | | | | | | | | | To a great extent |
| --- | --- | --- | --- | --- | --- | --- | --- | --- | --- | --- |
| Currently | 1 | 2 | 3 | 4 | 5 | 6 | 7 | 8 | 9 | 10 |
| Expectation after FRAMES | 1 | 2 | 3 | 4 | 5 | 6 | 7 | 8 | 9 | 10 |

What will be done in FRAMES with regard to flood preparation, that improves the

1.	physical resilience in the pilot area?

2.    capacities of local organisations/institutions in the pilot area?
3.    capacities of local communities (citizens, businesses) in the pilot area?

Appendix B.1.5 Flood Recovery

Facilitates a good and fast recovery after a flood event. Includes plans for draining/pumping away flood water and restoring safety and security, plans for reconstruction or rebuilding critical infrastructure, damage compensation/insurance systems, return of evacuated communities and social-psychological support.

To what extent is flood recovery a strong characteristic of the pilot area?

|  | Not at all |  |  |  |  |  |  |  |  | To a great extent |
|---|---|---|---|---|---|---|---|---|---|---|
| Currently | 1 | 2 | 3 | 4 | 5 | 6 | 7 | 8 | 9 | 10 |
| Expectation after FRAMES | 1 | 2 | 3 | 4 | 5 | 6 | 7 | 8 | 9 | 10 |

What will be done in FRAMES with regard to flood recovery, that improves the

1.    physical resilience in the pilot area?
2.    capacities of local organisations/institutions in the pilot area?
3.    capacities of local communities (citizens, businesses) in the pilot area?

*Appendix B.2 Resilience of Authorities and Communities*

Appendix B.2.1 Flood Resilience of Authorities

Please name the organisations/stakeholders that will be involved in your pilot:

| | **Organization** |
|---|---|
| 1 | |
| 2 | |
| 3 | |
| 4 | |
| 5 | |
| 6 | |
| 7 | |
| 8 | |
| 9 | |
| 10 | |

In general, to what extent is flood risk mitigation embedded in policy and practice of these organisations, in your opinion?

|  | Not at all |  |  |  |  |  |  |  |  | To a great extent |
|---|---|---|---|---|---|---|---|---|---|---|
| Currently | 1 | 2 | 3 | 4 | 5 | 6 | 7 | 8 | 9 | 10 |
| Expectation after FRAMES | 1 | 2 | 3 | 4 | 5 | 6 | 7 | 8 | 9 | 10 |

Please shortly explain your answer:

In general, to what extent is flood preparation embedded in policy and practice of these organisations, in your opinion?

|  | Not at all |  |  |  |  |  |  |  |  | To a great extent |
|---|---|---|---|---|---|---|---|---|---|---|
| Currently | 1 | 2 | 3 | 4 | 5 | 6 | 7 | 8 | 9 | 10 |
| Expectation after FRAMES | 1 | 2 | 3 | 4 | 5 | 6 | 7 | 8 | 9 | 10 |

Please shortly explain your answer:

In general, to what extent is flood recovery embedded in policy and practice of these organisations, in your opinion?

| | Not at all | | | | | | | | | To a great extent |
|---|---|---|---|---|---|---|---|---|---|---|
| Currently | 1 | 2 | 3 | 4 | 5 | 6 | 7 | 8 | 9 | 10 |
| Expectation after FRAMES | 1 | 2 | 3 | 4 | 5 | 6 | 7 | 8 | 9 | 10 |

Please shortly explain your answer:

Appendix B.2.2 Flood Resilience of Local Communities

Please name the communities (e.g., neighbourhoods, municipalities) that will be involved in/informed about your pilot(s), and how many citizens they consist of:

| | Community | Number of Citizens (Approximately) |
|---|---|---|
| 1 | | |
| 2 | | |
| 3 | | |
| 4 | | |
| 5 | | |
| 6 | | |
| 7 | | |
| 8 | | |
| 9 | | |
| 10 | | |

In general, to what extent is flood risk mitigation embedded in the behaviour of these communities, in your opinion?

| | Not at all | | | | | | | | | To a great extent |
|---|---|---|---|---|---|---|---|---|---|---|
| Currently | 1 | 2 | 3 | 4 | 5 | 6 | 7 | 8 | 9 | 10 |
| Expectation after FRAMES | 1 | 2 | 3 | 4 | 5 | 6 | 7 | 8 | 9 | 10 |

Please shortly explain your answer:

In general, to what extent is flood preparation embedded in the behaviour of these communities, in your opinion?

| | Not at all | | | | | | | | | To a great extent |
|---|---|---|---|---|---|---|---|---|---|---|
| Currently | 1 | 2 | 3 | 4 | 5 | 6 | 7 | 8 | 9 | 10 |
| Expectation after FRAMES | 1 | 2 | 3 | 4 | 5 | 6 | 7 | 8 | 9 | 10 |

Please shortly explain your answer:

In general, to what extent is flood recovery embedded in the behaviour of these communities, in your opinion?

| | Not at all | | | | | | | | | To a great extent |
|---|---|---|---|---|---|---|---|---|---|---|
| Currently | 1 | 2 | 3 | 4 | 5 | 6 | 7 | 8 | 9 | 10 |
| Expectation after FRAMES | 1 | 2 | 3 | 4 | 5 | 6 | 7 | 8 | 9 | 10 |

Please shortly explain your answer:

**Appendix C  Interview Guideline**

The interview guideline was developed between December 2018 and January 2019 by 7 project partners from the knowledge institutes: 4 from HZ Delta Academy, 1 from Oldenburg University and 2 from Ghent University.

General information
Pilot area:
Name of the pilot manager:
Organization:
Interviewers:
Date:
Objective:

This interview consists of 14 semi-structured questions to facilitate/guide an open discussion with the pilot coordinators/managers about the implementation process of the MLS approach in the pilot area and the impact/influence of the expected outcomes in the pilot region. This input will be used for the development of the FRAMES Decision Support System and resilience toolkit. The bullet lists relevant aspects to follow and bring into the discussion if they are mentioned by the pilot manager.

Introductory questions:

1.  Which were/are the most relevant flood risk management measures taken in the pilot area and in the pilot region (*before* FRAMES?)?
2.  How did you get involved in setting up this pilot and what was your main motivation?

Key questions:

3.  Main goals/MLS: What are the main goal(s) in the pilot in relation to the MLS layers (protection/prevention/preparation/recovery)?

    ✓     Specify if the goal(s) is/are for short/medium/long term (<5/5–25/>25 years)
    ✓     Reason of the pilot, was it completely new compared to the region's FRM approach?

4.  Pilot process and decision making: Throughout the implementation process of MLS, which were/are

    ✓     Follow up steps towards new/improved strategies in FRM
    ✓     Tools/methods used to perform and monitor these steps/decisions
    ✓     Current and missing actors (level, sector)
    ✓     Tools/methods to involved and keep actors engaged (special attention of local communities)

5.  Interaction of layers/hierarchy of layers: Have you encountered any interaction of activities between several layers of MLS? If yes, how have you coordinated activities when they take place on different layers? Could you give any examples?
6.  Struggles/hurdles: Which were/are the main struggles that you have faced in implementing and reaching the goal(s) in relation to the MLS approach? What would you never do again? How have you overcome these struggles?

    ✓     Drivers/barriers of change: stakeholders, time, resources, uncertainties of climate change, power

7.  Accomplishments: What did it go well? Would you do again the same thing? If not, what would you change?
8.  (Expected) Outcomes: Which are the outcomes or expected outcomes of implementing the MLS in the pilot area and in the pilot region?

    ✓    Concrete outputs/products (new or not?)

    ✓    Share information/communicate outcomes

9.    Embedding/upscaling of MLS approach. Based on the pilot (expected) outcomes, what are the potential/opportunities to embed/upscale the MLS approach in the region after FRAMES? For example, community resilience, spatial planning/infrastructure, FRM policies/strategies.

10.    Role of pilot coordinator: How do you see your role as pilot manager in facilitating the implementation of the MLS approach during the pilot project and embedding/upscaling the project outcomes *afterwards*?

11.    Role of key actors: How do you see the role of the key actors in facilitating the implementation of the MLS approach *during* the pilot project and embedding/upscaling the project outcomes *afterwards*?

12.    What do you think will be the main drivers/barriers for the embedding/upscaling of the expected outcomes after FRAMES?

13.    What is needed to make use of these drivers and to overcome these barriers?

Ending questions:

14.    Do you recommend to talk with the key actors that will have a role in enabling/facilitating the implementation of MLS approach further, after FRAMES, at regional/national level?

15.    Do you have any comments related to this interview? (additional questions or ideas).

Thanks so much for your time and contribution in this interview

## Appendix D Transnational Focus Groups (TFG)

Table A3 lists the 12 FRM actions derived from the 15 pilot projects of FRAMES.

**Table A3.** List of FRM actions, derived from pilot activities.

| No | Flood Resilience Actions in Pilot Project |
|----|-------------------------------------------|
| 1 | *Improving zoning of developments in flood prone areas**  |
| 2 | Reducing surface flood risk from extreme rainfall via increasing storage capacity in private and public space |
| 3 | Lowering flood risk for communities via nature based solutions upstream |
| 4 | 4a Realizing a flood proof critical infrastructure<br>4b Limit cascade-effects of critical infrastructure failure |
| 5 | *Integrate emergency response planning in flood risk management (and vice versa) *  |
| 6 | Improve strategies for preventive evacuation |
| 7 | Develop alternative evacuation strategies (safe haven, shelters, vertical evacuation) |
| 8 | Raising awareness for flood resilience measures |
| 9 | Involving communities in flood resilience measures |
| 10 | 10a Empower individuals and business to take measures themselves (self-reliance)<br>10b *Empower communities to take action for local flood resilience measures *  |
| 11 | Support local authorities and communities in adaptive planning for flood resilience |
| 12 | Apply adaptive planning to look for tipping points in flood risk management strategies and to explore synergetic combinations of strategies |

* Actions used during the TFGs discussions.

The total number of 31 project partners participated during of the 3 TFGs held on 27th March 2019 in Oldenburg: 12 from The Netherlands (3/HZ Delta Academy, 4/Province of Zeeland, 3/Province of

South Holland, 1/Rijkswaterstaat and 1/Reeleaf); 8 from Germany (2/Jade UAS, 3/Oldenburg University, 1/Küste & Raum Consultancy, 1/Oldenburg-East-Frisian Water Board and 1/Jade HS-citizens science-); 1 from Denmark (1/Danish Coastal Authority); 3 from Belgium (1/Ghent University, 2/Provincie Oost-Vlaanderen); and 7 from UK (1/Kent County Council, 2/National Flood Forum, 1/Trent River Trust, 2/South East River Trust and 1/Tees Rivers Trust).

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
