# Peer review of "Adaptive Capacities for Diversified Flood Risk Management Strategies: Learning from Pilot Projects"

_water, doi:10.3390/w11122643_

Round 1
Reviewer 1 Report
This paper presents an analytical framework using the Adaptive Capacity Wheel and Triple loop learning to evaluate the adaptive capacities needed for diversified flood risk management (FRM). To do so it focuses on two FRM pilot projects in the Netherlands and Germany and the types of adaptive capacities that were missing, employed and developed through these projects. This paper is well written and is relevant to this journal and of interest to its readers. Overall this is a good and interesting paper, but there are some areas that would benefit from being improved. However, there is an important caveat which relates to the methodology used and in particular the participants involved in the surveys and interviews – the authors need to clarify their methodology for us to better understand the diversity of stakeholders surveyed/interviewed in this study as for now it seems that the results of this study only reflect the perspectives of the project managers of each pilot, which would be of real concern to the value and validity of this study.
The methods section needs to be improved to provide greater clarity on a few issues:
Could the authors please clarify whether the questionnaire provided in Annex A is just a sample of the full questionnaire or the whole questionnaire. In section 4.2 the authors mention that the pilot managers had to respond to questions for the current and expected situation after the pilot project, but the questionnaire provided in Annex A only shows questions for the current situation and not for the expected situation. However, it could be that the questionnaire provided in Annex A is not the full questionnaire in which case this is fine, but should be made clear. It is also not clear how many people responded to the questionnaire – the authors state that for each pilot project a questionnaire was completed by the pilot manager, so does this mean that only two questionnaires were completed in total? One for the pilot in Germany and one for the pilot in The Netherlands? Or is it for the 15 pilot projects across the five countries? And for each questionnaire did only one person reply (e.g. the project manager)? If this is the case then this seems like a strong reliance on one individual’s perceptions/understanding of the situation in each case study. Could the authors please better explain how the questionnaire was administered and how many people completed it. In the questionnaire provided in Appendix A it appears that multiple people may contribute to completing the questionnaire but if that is the case did they each provide an answer to all questions or did they have to come to an agreement to provide only one answer per question? And if that is the case how was this process managed? This is quite critical as it will help better understand the robustness of the findings from the authors. Authors also need to better explain how many people were interviewed. They state that all pilot managers were interviewed and two interviews were used for the selected pilot projects, so does that mean that 17 interviews were done? And who were the extra interviews for the selected pilot projects with? Or were these follow up interviews? In addition, if the surveys and interviews were all done only with the pilot project managers it does give this study a strong reliance/dependence on the opinion of only person per pilot, which would be a real concern. It could be that other people involved in the projects have very different perspectives from the project managers – how was this taken into account in this study to ensure that the findings from the study accurately reflect what was done in the pilots and the opinions/perspectives of all stakeholders involved in the pilot projects?
In addition, this paper would benefit from a section presenting how flood risk management has historically and is currently being done in Germany and The Netherlands, as this is in many ways critical to understanding what is being done in the pilots and why, and the barriers they face – eg strong emphasis on protection making spatial planning ‘less’ relevant. The political economy of FRM in these two countries is important to this study and needs to be brought out more clearly. This could also help strengthen the discussion section regarding the relevance of this study more broadly when it comes to improving flood risk management across Europe and improving projects being funded to support FRM in Europe. If projects are supposed to lead to change/improved policies or testing ground for new ideas/innovations that can then be scaled out – what recommendations come from this study on this? I feel the authors could say more on this then they are doing at present. What is the wider relevance of this study for improved or more transformative FRM in Europe?
Also, do the authors see the transferability of their framework/approach to other pilots/projects not related to FRMs (so obviously the categories such as mitigation, preparedness, etc would be different) to better understand the impact of those pilots/projects and whether they can lead to change/transformation and to wider scale out?
How to scale out local projects/pilots is an issue of great interest within current research on sustainable development, climate change adaptation, climate resilient development etc so it would be good if the authors could engage in this discussion a little.
Minor comments:
Table 5 – legend mentions shaded capacities but not sure what these refer to in the table – this is not clear.
Author Response
Rebuttal for manuscript ID: water-618562
Title: Adaptive capacities for diversified flood risk management strategies: learning from pilots
Authors: Flavia Simona Cosoveanu, Jean-Marie Buijs, Marloes Bakker, Teun Terpstra
Dear editors,
We thank the reviewers for their insightful comments on our manuscript. We highly appreciate the constructive feedback, suggestions for literature, textual improvements and the discussion (all reviewers). Reviewer 1 and 3 asked relevant questions to clarify the description of the methodology concerning the process of administering the questionnaires and number of interviews. Also relevant feedback has been provided about adaptive capacities (reviewer 2) and the interpretation of different learning loops (reviewer 3 and 4).
Below we have listed all the comments and suggestions per reviewer. After each comment we have explained our response, provided clarification where needed and where and how we have processed this with ‘Track changes’(All Markup) in the word version of the paper.
We are convinced that the reviews have helped us to substantially improve the paper. We are looking forward to the review of the revised manuscript.
Looking forward to hear from you
Kind regards
Flavia, Jean-Marie, Marloes and Teun
Responses to reviewer 1
1. However, there is an important caveat which relates to the methodology used and in particular the participants involved in the surveys and interviews – the authors need to clarify their methodology for us to better understand the diversity of stakeholders surveyed/interviewed in this study as for now it seems that the results of this study only reflect the perspectives of the project managers of each pilot, which would be of real concern to the value and validity of this study. The methods section needs to be improved to provide greater clarity on a few issues:
a) Could the authors please clarify whether the questionnaire provided in Annex A is just a sample of the full questionnaire or the whole questionnaire. In section 4.2 the authors mention that the pilot managers had to respond to questions for the current and expected situation after the pilot project, but the questionnaire provided in Annex A only shows questions for the current situation and not for the expected situation. However, it could be that the questionnaire provided in Annex A is not the full questionnaire in which case this is fine, but should be made clear.
The questionnaire provided in Appendix B (previously Appendix A) is the full questionnaire as sent out to the pilot managers. Since changes to strategies of pro-action and protection were not part of the pilot projects, we did not ask for the expected situation for these strategies. For other strategies and aspects of flood resilience we have asked the respondents to answer the questions both for the current situation and expected situation (see Questionnaire template, Appendix B, Question 1.3 ~ 2.2).
b) It is also not clear how many people responded to the questionnaire – the authors state that for each pilot project a questionnaire was completed by the pilot manager, so does this mean that only two questionnaires were completed in total? One for the pilot in Germany and one for the pilot in The Netherlands? Or is it for the 15 pilot projects across the five countries? And for each questionnaire did only one person reply (e.g. the project manager)? If this is the case then this seems like a strong reliance on one individual’s perceptions/understanding of the situation in each case study.
The questionnaire has been completed by the pilot managers of all 16 pilots in the five countries. We have instructed the pilot managers to distribute the questionnaire to key stakeholders. The results were collected by the authors For this paper we have only used the questionnaires of the selected cases. Each of these questionnaires have been completed by the pilot manager and other key stakeholders in the pilot project.
For the case Alblasserwaard, the questionnaire has been completed by 4 relevant stakeholders: Provincie Zuid-Holland (the pilot manage and policy advisor), Rijkswaterstaat (policy advisor), Regio Alblasserwaard en Vijfheerenlanden (policy advisor) and Noordelijjke Drechtsteden (policy advisor). In Wesermarsch the questionnaire has been completed by the Jade University as pilot manager (full professor and researcher) Landkreis Wesermarsch (head of crisis management at the County) Wesermarsch, Küste & Raum (Independent Consultant) and drinking water company OOWV (spatial planning manager, strategic planning manager, consultant innovation networks). Since the main focus of the paper is to have insight in the process of the pilot projects, we have included the aggregated results in the paper.
In the manuscript this process has been more clearly described – see lines 198-207.
c) Could the authors please better explain how the questionnaire was administered and how many people completed it. In the questionnaire provided in Appendix A it appears that multiple people may contribute to completing the questionnaire but if that is the case did they each provide an answer to all questions or did they have to come to an agreement to provide only one answer per question? And if that is the case how was this process managed? This is quite critical as it will help better understand the robustness of the findings from the authors.
This issue relates strongly with 1b. The observation by the reviewer that multiple people have contributed to completing the questionnaire is correct and needed more explanation. We have instructed the pilot managers about the procedure of the questionnaire and they distributed this to key actors. Each actor completed the questionnaire separately. For the aim of the paper we have included that aggregated results of the questionnaire, by averaging the scores of the closed questions and bringing together all answers on the open questions. We have explained this process in the manuscript, lines 187-196.
2. Authors also need to better explain how many people were interviewed. They state that all pilot managers were interviewed and two interviews were used for the selected pilot projects, so does that mean that 17 interviews were done? And who were the extra interviews for the selected pilot projects with? Or were these follow up interviews? In addition, if the surveys and interviews were all done only with the pilot project managers it does give this study a strong reliance/dependence on the opinion of only person per pilot, which would be a real concern. It could be that other people involved in the projects have very different perspectives from the project managers – how was this taken into account in this study to ensure that the findings from the study accurately reflect what was done in the pilots and the opinions/perspectives of all stakeholders involved in the pilot projects?
The observation by the reviewer that we interviewed the pilot managers of 16 pilots in total is correct. For the selected cases, we only included the interview results with these pilot managers. We agree with the reviewer that it is relevant to involve people from different perspectives in case studies. As argued in relation to issue 1, this has been taken into account in the questionnaire, which has been completed by multiple actors in the pilot project (see 1b). The interviews with all pilot managers contributed to developing the framework. In this paper we test this framework by identification of missing, employed and developed capacities in two pilot projects. We agree with the reviewer that having interviews with more stakeholders about adaptive capacities would be of added value for the case studies. Since the focus of this paper is on the development of the framework, we see this as a limitation of the current paper. In the methodology of this paper we have mitigated this via the questionnaire, which has been completed by multiple stakeholders (7 in Wesermarsch and 4 in Alblasserwaard), and the three focus groups with approx. 30 FRM professionals.
This process is described in section 4.2 to 4.4. and 4.6. The limitations of the methodology are discussed in section 6, see lines 615-618.
3. In addition, this paper would benefit from a section presenting how flood risk management has historically and is currently being done in Germany and The Netherlands, as this is in many ways critical to understanding what is being done in the pilots and why, and the barriers they face – eg strong emphasis on protection making spatial planning ‘less’ relevant.
We agree with the reviewer that the historical and current flood risk management approaches in both countries are critical to understand the current pilots. For this reason we have included the situation ‘before’ each pilot throughout our analysis. In the introduction of both cases we already highlighted the main flood risk governance characteristics of both countries (Table 2). Since several other authors have described the historical characteristics in more detail, we have decided to link this section better to the existing literature (see lines 177-183).
4. The political economy of FRM in these two countries is important to this study and needs to be brought out more clearly. This could also help strengthen the discussion section regarding the relevance of this study more broadly when it comes to improving flood risk management across Europe and improving projects being funded to support FRM in Europe. If projects are supposed to lead to change/improved policies or testing ground for new ideas/innovations that can then be scaled out – what recommendations come from this study on this? I feel the authors could say more on this then they are doing at present. What is the wider relevance of this study for improved or more transformative FRM in Europe?
We see the relevance to include the political economy of FRM and link this to FRM across Europe by looking into the supposed change and scaling of pilots. We have addressed this in the Conclusion (lines 607-612) by referring to the ‘pilot paradox’ as a valuable approach for this process. However, in this paper we focused on the development and testing of the framework in relation to two pilots. This requires also some modesty to provide more generic recommendations. We feel this topic is beyond the scope of this paper.
5. Also, do the authors see the transferability of their framework/approach to other pilots/projects not related to FRMs (so obviously the categories such as mitigation, preparedness, etc would be different) to better understand the impact of those pilots/projects and whether they can lead to change/transformation and to wider scale out?
In the conclusion it has been brought forward that the findings contribute to theories about niche – regimes interactions [20] and policy transfer via pilot projects (lines 598-599) and we suggest that the use of the ‘pilot paradox’ approach for further research can result in more knowledge about how to upscale pilot projects (lines 607-612). In this perspective we see the potential transferability to other domains, but we also feel the framework needs to be validated more on FRM projects before we can provide more recommendations of other applications.
6. How to scale out local projects/pilots is an issue of great interest within current research on sustainable development, climate change adaptation, climate resilient development etc so it would be good if the authors could engage in this discussion a little.
This comments is linked to the previous comment. We have limited ourselves in the concluding sections to the answer provides to comment 5 to remain focus on the framework and what can be concluded about diversification of flood risk management.
7. Table 5 – legend mentions shaded capacities but not sure what these refer to in the table – this is not clear.
The legend of table 5 has been changed: shaded has been replaced with ‘grey’ color to clarify the meaning of the ‘shading’ which refers to the lacking capacities before the pilot project that result in developed capacities at the end of the pilot project.
Reviewer 2 Report
This is a well-researched paper on an important subject but suffers from a number of minor – but detracting – issues that deter it from being publishable in its present form. With revision and bolstering of some sections of the paper, it can become a significant contribution to the flood risk mitigation-adaptive governance debate. Here are my enumerated edits and comments:
Line 31 – climate change not global warming
Line 39 – some enumeration here – or elsewhere in the manuscript – of specific adaptive measures should be offered; e.g., low impact measures such as retreat from the shoreline; reclaiming shoreline from the built environment.
Line 84 - Other work has been done in the North American and European context with respect to water governance reform whose lessons could be useful – e.g., institutional change and possibilities for cross-jurisdictional cooperation; legal reform, etc. See, e.g.
1. Huitema, D., Mostert, E., Egas, W., Moellenkamp, S., and Pahl-Wostl, C. (2009) “Adaptive Water Governance: Assessing the Institutional Prescriptions of Adaptive Co-Management from a Governance Perspective and Defining a Research Agenda.” Ecology and Society 14;
2. Kallis G, Kiparsky M, Norgarrd R. “Collaborative governance and adaptive management: lessons from California’s CALFED Water Program.” Environ Sci Policy 2009, 12:631–643. doi: 10.1016/j.envsci.2009.07.002.
Line 86 – Distinction between quantitative and qualitative adaptive capacities is unclear. What does this mean – measureable vs. non-measureable?
Table 1 – Another adaptive capacity cited in the literature is boundary spanning – the ability to span different domains and to think cross-disciplinarily.
Line 102 – Some discussion of how learning outcomes could also include the extent to which the problem itself is defined is needed – e.g., is flooding and its damages to the built environment, or are there greater risks to the environment in fortifying against flooding?
Line 133 – change to “The key question is whether developed adaptive capacities result….”
Line 141 – change to “Which adaptive capacities were determined to be missing before the pilot project?” Logically, how can you know what was missing until you first undertake the project?
Lines 141, 155 – Not necessary to say empirical data – all data are derived empirically: just “data.”
Line 185 – “In total,” not “So in total….”
Line 202 – delete “implicitly.”
Line 209 – “content verified”, not “checked” (replace text).
Line 236 – add: projects are located in….
Lines 241-243 – pilot projects not pilots projects.
Line 243 and throughout the paper – spelling should be focused not focussed.
Line 245 – would, not will increase.
Line 260- replace lacking capacities with “capacities needed for upscaling that were not present.”
Line 263ff – both cases studies should be rewritten as a simple narrative structure, not as a series of bulletized headings. These make the manuscript read more like a report than a scholarly article.
Line 344 – instead of diversity of solutions, I think what’s being argued for is variety of solutions – you want options that vary across structural/non-structural, and fitted for meeting different objectives.
Line 473 – This discussion needs to consider what specific information needs are implied by this finding: visualization of flood risks? Likely flood depths; resources available for planning? Provide some detail.
Line 514 – this is where the Conclusion begins, and should have a new sub-heading here. Also change line 435 – to Discussion.
Line 518 – should be rewritten to state: “The findings contribute . . . “
Line 528 – rephrase as: “Here we mention two avenues needing improvement.”

Author Response
Rebuttal for manuscript ID: water-618562
Title: Adaptive capacities for diversified flood risk management strategies: learning from pilots
Authors: Flavia Simona Cosoveanu, Jean-Marie Buijs, Marloes Bakker, Teun Terpstra
Dear editors,
We thank the reviewers for their insightful comments on our manuscript. We highly appreciate the constructive feedback, suggestions for literature, textual improvements and the discussion (all reviewers). Reviewer 1 and 3 asked relevant questions to clarify the description of the methodology concerning the process of administering the questionnaires and number of interviews. Also relevant feedback has been provided about adaptive capacities (reviewer 2) and the interpretation of different learning loops (reviewer 3 and 4).
Below we have listed all the comments and suggestions per reviewer. After each comment we have explained our response, provided clarification where needed and where and how we have processed this with ‘Track changes’(All Markup) in the word version of the paper.
We are convinced that the reviews have helped us to substantially improve the paper. We are looking forward to the review of the revised manuscript.
Looking forward to hear from you
Kind regards
Flavia, Jean-Marie, Marloes and Teun
Responses to reviewer 2
Line 31 – This has been changed, as suggested.
Line 39 – As suggested, specific adaptive measures are now provided
Line 84 - The suggested references are relevant literature suggestions. In the section to which the reviewers refer, we concentrate on the Adaptive Capacity Wheel of Gupta et al (2010). The suggested literature discusses the need of collaborative networks for adaptive (water) management. Int he structure of our paper, this fits best in the Discussion section where the capacity collaborative leadership is discussed. We have included this in lines 545-548
Line 86- This has now been explained in more detail- see lines 91-95
Table 1- We agree with the reviewer that boundary spanning is relevant to discuss in relation to adaptive capacities for diversifying FRMS. In the framework applied in this paper we consider boundary spanning capacities as part of the capacity collaborative leadership. We have explained this in the Discussion: 'Alignment across sectoral doundaries is key in governance arrangements for adapting to climate change, which is also observed in both cases. Boundary spanning interactions, including cherishing boundaries for clear allocation of responsibilities, is required for collective action in diversifying FRMS.' (lines 548-551)
Line 102 - We have included also examples for single, double and triple loop learning, linked to the topic of FRM. See lines 105, 108-110, 113-115. In the discussion we have added reframing of the problem to the discussion about reframing considering long term processes such as soil subsidence and sea level rise (lines 592-593)
Line 133 - This has been changed, as suggested, now line 144.
Lines 141 - This has been changed, as suggested. Now line 151.
Lines 141, 155- This has been changed, as suggested. Now line 159.
Line 184 - This has been changed, as suggested in line 213 and it was replaced by ‘Thus, overall..’ in line 226.
Line 201 - This has been changed, as suggested in line 231.
Line 207 - The term ‘checked’ was replaced by ‘reviewed’, see line 276.
Line 236 - This has been changed, as suggested, now line 265.
Lines 241-241 This has been changed, as suggested, now line 270-271.
Lines 243 and throughout the paper - This has been changed, as suggested throughout the whole paper.
Line 244 - This has been changed, as suggested in line 274.
Line 260 - This has been changed, as suggested, now in line 290-291.
Line 263 - Although this issue was only brought forward by this reviewer, we agree that a simple narrative structure matches better with the format of a scholarly article. We have rewritten this part of the. See section 5.2.1 and 5.2.2
Line 344 - As suggested, diversity has been replaced with variety in line 391.
Line 473 - Examples of type of information needs are specified in lines 540-544: ‘’For instance, step by step checklist for farmers to prepare themselves and their livestock (Wesermarsch) and how entrepreneurs can protect their businesses (Alblasserwaard-Vijfheerenlanden) in case of flooding. Making the information on emergency planning available for the actors resulted in enhanced mutual understanding of interests, actions and needs’.
Line 511 - As suggested, Conclusion starts now in line 594.
Line 514 - This has been changed, as suggested, now in line 598
Line 528 - This has been changed, as suggested in line 607.
Reviewer 3 Report
1) The criteria for choosing the two case studies is not sufficiently clear. A summary table of the key adaptive characteristics for all of the pilot studies would be useful so that the reader can see why some pilots were included and others rejected.
2) The attachment of the appendices is useful however elements of these should be incorporated in the body of the text of the paper. This will contribute to the understanding of the relationship of the empirical data and theoretical/conceptual framework.
3) The single loop learning as described here is essentially an incremental rational process for learning and deciding how project actions or planned set of actions evolves through the implementation of the pilot project. This approach is very much similar to the theory of planning advanced by Charles Lindblom (1959). I have attached two URL sources that may provide a more complete description of what has become the single loop frame for understanding this type of project management.

Author Response
Rebuttal for manuscript ID: water-618562
Title: Adaptive capacities for diversified flood risk management strategies: learning from pilots
Authors: Flavia Simona Cosoveanu, Jean-Marie Buijs, Marloes Bakker, Teun Terpstra
Dear editors,
We thank the reviewers for their insightful comments on our manuscript. We highly appreciate the constructive feedback, suggestions for literature, textual improvements and the discussion (all reviewers). Reviewer 1 and 3 asked relevant questions to clarify the description of the methodology concerning the process of administering the questionnaires and number of interviews. Also relevant feedback has been provided about adaptive capacities (reviewer 2) and the interpretation of different learning loops (reviewer 3 and 4).
Below we have listed all the comments and suggestions per reviewer. After each comment we have explained our response, provided clarification where needed and where and how we have processed this with ‘Track changes’(All Markup) in the word version of the paper.
We are convinced that the reviews have helped us to substantially improve the paper. We are looking forward to the review of the revised manuscript.
Looking forward to hear from you
Kind regards
Flavia, Jean-Marie, Marloes and Teun
Responses to reviewer 3
1. The criteria for choosing the two case studies is not sufficiently clear. A summary table of the key adaptive characteristics for all of the pilot studies would be useful so that the reader can see why some pilots were included and others rejected.
We agree with the reviewer that the criteria were not specified in a clear manner. We have therefore added a Table A1 in Appendix with the main flood risk management actions of all the FRAMES pilot projects. For a description of all processes of pilot projects, see www.frameswiki.eu
The selection of the Alblasserwaard and the Wesermarsch pilots is based first on the traditional flood management that mainly relies on flood defense with hard infrastructure. (see Table 2 and lines 176-193). In addition, the goal of both pilots is to diversify flood risk management by taking actions in flood preparedness, flood evacuation and spatial adaptation (see Table 2 and lines 164-175). The selection of the two cases for this paper is based on the intended integration of Mitigation (via spatial planning) and Preparedness (awareness raising and evacuation planning). Besides the two selected cases also the Sloegebied pilot intended to focus at the same aspects, but this pilot was not finished yet when writing this paper. The table has been added in Appendix A to explain the selection of the Alblasserwaard and Wesermarsch as case studies. This has been referred to in lines 178-179.
2. The attachment of the appendices is useful however elements of these should be incorporated in the body of the text of the paper. This will contribute to the understanding of the relationship of the empirical data and theoretical/conceptual framework.
We consider that the survey (Appendix B) and the interview guideline (Appendix C) are too long to include in the text of the paper. Examples of the questions as part of the questionnaire and the interview are mentioned in the main text (lines 200-204, 208-228 and lines 233-235, respectively). Likewise, the list of FRM actions and workshop participants used during the TFG (the Appendix D) was not included in the main text because it’s too detailed information. Only the actions used during the TFG are mentioned in the main text (lines 250-252).
3. The single loop learning as described here is essentially an incremental rational process for learning and deciding how project actions or planned set of actions evolves through the implementation of the pilot project. This approach is very much similar to the theory of planning advanced by Charles Lindblom (1959). I have attached two URL sources that may provide a more complete description of what has become the single loop frame for understanding this type of project management.
Unfortunately the two URL sources were not included in the version of the review we received. However, we consider the suggested theory of Lindblom and provided explanation relevant. We have included an explanation of this, based on planning literature, in the Discussion: ‘This aligns with planning literature, which emphasizes that planning practices are more adaptive (adjust to changing circumstances) and incremental (gradual changes) than often assumed by scholars proposing ‘new’ planning approaches’ (see lines 572-574).
Reviewer 4 Report
This is an outstanding idea, presented in an interesting but incomplete way. One missing item, that the authors may believe is outside their purview is that a significant part of FRMS is spatial planning at the catchment scale. That would be relevant to their objectives and to decisions made, but perhaps outside their present paper.
More importantly, they present loops (i.e., single, double, triple) as an innovative way to think about FRMS. Then they demonstrate some things about single and double loop learning. Triple loop learning is briefly introduced, then they only briefly comment even on the double loop learning from the pilots. They apparently did not address triple loop learning at all. I think they should either get that out of the introduction or interpret the results in that light.

Author Response
Rebuttal for manuscript ID: water-618562
Title: Adaptive capacities for diversified flood risk management strategies: learning from pilots
Authors: Flavia Simona Cosoveanu, Jean-Marie Buijs, Marloes Bakker, Teun Terpstra
Dear editors,
We thank the reviewers for their insightful comments on our manuscript. We highly appreciate the constructive feedback, suggestions for literature, textual improvements and the discussion (all reviewers). Reviewer 1 and 3 asked relevant questions to clarify the description of the methodology concerning the process of administering the questionnaires and number of interviews. Also relevant feedback has been provided about adaptive capacities (reviewer 2) and the interpretation of different learning loops (reviewer 3 and 4).
Below we have listed all the comments and suggestions per reviewer. After each comment we have explained our response, provided clarification where needed and where and how we have processed this with ‘Track changes’(All Markup) in the word version of the paper.
We are convinced that the reviews have helped us to substantially improve the paper. We are looking forward to the review of the revised manuscript.
Looking forward to hear from you
Kind regards
Flavia, Jean-Marie, Marloes and Teun
Responses to reviewer 4
1. One missing item, that the authors may believe is outside their purview is that a significant part of FRMS is spatial planning at the catchment scale. That would be relevant to their objectives and to decisions made, but perhaps outside their present paper.
The Alblasserwaard pilot is located downstream the Rhine river and the Wesermarsch in a coastal area of the North Sea Region. Both locations are flood prone areas. Therefore, we agree with the fact that spatial planning is an approach that reduces flood risk at river catchment. Spatial planning at catchment level was not taken into account in the pilot studies which were selected as case studies for this paper. We see the relevance of this for both flood prone areas, but this would also require reframing (double loop learning) of FRMS problems and objectives. We link to this in the Discussion, lines 518-535.
2. More importantly, they present loops (i.e., single, double, triple) as an innovative way to think about FRMS. Then they demonstrate some things about single and double loop learning. Triple loop learning is briefly introduced, then they only briefly comment even on the double loop learning from the pilots. They apparently did not address triple loop learning at all. I think they should either get that out of the introduction or interpret the results in that light.
We agree with the reviewer that triple loop learning is not addressed in the Results section of the case studies. Considering the topic of diversified flood risk management strategies and long term challenges of soil subsidence and sea level rise for this field, we consider the idea of triple loop learning relevant for the framework. We have reflected briefly on this in the discussion:
‘Besides more urgency [50], also a different type of pilot projects are needed to deal with long term processes such as soil subsidence and sea level rise’. (see lines 592-593)
Round 2
Reviewer 1 Report
The authors have addressed my main concerns regarding the methodology used in this research. In addition, they have addressed the remaining comments I had and clarified that the focus of the paper is very much on presenting an analytical framework, which is why the conclusion focuses on the value of the framework itself. I have no additional comments and am happy for this paper to be published.
Author Response
Dear reviewer
We appreciate very much the time spent in reviewing our paper twice. The comments provided were very beneficial to improve and clarify the aim of the framework in the paper. Thus, we have addressed all comment with great attention. Thank you for accepting the publication of this paper.
With kind regards
Flavia, Jean-Marie, Teun and Marloes